# The Mechanism of Short-Circuit Oscillations in Automotive-Grade Multi-Chip Parallel Power Modules and an Effective Mitigation Approach

**DOI:** 10.3390/s24092858

**Published:** 2024-04-30

**Authors:** Kun Ma, Yameng Sun, Xun Liu, Yifan Song, Xuehan Li, Huimin Shi, Zheng Feng, Xiao Zhang, Yang Zhou, Sheng Liu

**Affiliations:** 1Institute of Technological Sciences, Wuhan University, Wuhan 430072, China; mk_175@whu.edu.cn (K.M.); sunyameng@whu.edu.cn (Y.S.); xun.liu@whu.edu.cn (X.L.); xiao.zhang@whu.edu.cn (X.Z.); 2School of Mechanical Science and Engineering, Huazhong University of Science and Technology, Wuhan 430070, China; syfmvp@163.com (Y.S.); xhli2022@outlook.com (X.L.); 3Hefei Archimedes Electronic Technology Co., Ltd., Hefei 230000, China; 4School of Power and Mechanical Engineering, Wuhan University, Wuhan 430072, China; 5School of Microelectronics, Wuhan University, Wuhan 430072, China

**Keywords:** gate oscillation, multi-chip parallel parasitic parameters, power module, short-circuit

## Abstract

This paper presents an in-depth analysis of the oscillation phenomenon occurring in multi-chip parallel automotive-grade power modules under short-circuit conditions and investigates three suppression methods. We tested and analyzed two commercial automotive-grade power modules, one containing two chips and the other containing a single chip, and found that short-circuit gate oscillations were more likely to occur in multi-chip parallel packaged modules than in single-chip packaged modules. Through experimental and simulation analyses, we observed that gate oscillations were mainly caused by the interaction between internal parasitic parameters of the module and the external drive circuit, and we found that high drive resistance and low common emitter inductance between parallel chips could effectively suppress gate voltage oscillations. We also analyzed the two mainstream suppression schemes, increasing the drive gate resistance and placing the drive capacitors in parallel. Unfortunately, we found that these suppression schemes were not ideal solutions because both schemes changed the switching characteristics of the power module. As an alternative, we propose a simple and effective solution that involves adding parallel connections between the parallel chips. Simulation calculations showed that this optimized method reduced the emitter inductance between parallel chips in the upper bridge arm by about 30% and in the lower bridge arm by 35%. Through short-circuit experiments conducted at different DC bus voltages, it has been verified that the new optimized solution effectively resolves gate oscillation issues without affecting the switching characteristics of the power module.

## 1. Introduction

With the rise of Electric Vehicles (EVs) and Hybrid Electric Vehicles (HEVs), the demand for efficient, high-density power electronic components is increasing. Therefore, research and development in multi-chip parallel automotive-grade power modules has become particularly important. These modules play a key role in enhancing power density and system efficiency and must meet strict automotive safety and performance standards.

During the lifespan of automotive power modules, various extreme operating conditions may arise due to control unit failures, human errors, or other uncontrollable factors. One of the most common scenarios for automotive power modules is a short circuit (SC) [1,2,3]. To prevent module or power system damage under SC conditions, insulated-gate bipolar transistors (IGBTs) are designed to withstand high current and voltage conditions for a few microseconds (typically 10 μs), allowing the gate driver to shut down the IGBT before destruction [4]. Under certain operating conditions, high-frequency gate-emitter voltage oscillations (i.e., tens of megahertz) have been observed in both type 1 (turn-on transient) and type 2 (steady-state) SC conditions, and these oscillations can be a major problem in automotive-grade multi-chip parallel power modules [5]. If the oscillation amplitude exceeds the maximum gate voltage provided in the manufacturer’s datasheet, it may lead to gate oxide breakdown, permanently damaging the IGBT and severely limiting the SC robustness [6,7]. These oscillations can also degrade module lifespans and cause system reliability issues due to false turn-ons, additional losses, and electromagnetic interference.

Among the different IGBT failure mechanisms under SC conditions, high-frequency gate voltage oscillations have been reported with various interpretations. Compared to single-chip modules, SC gate oscillations are more likely to occur in multi-chip parallel modules [8,9]. It is well known that IGBT chips are inherently unstable at high collector voltages and high temperatures due to the presence of negative gate capacitance [10,11,12,13], and efforts have been made to improve layout designs to suppress these oscillations [8,14,15]. Another possible explanation is that the oscillations are initiated by the IGBT chip and enhanced by the internal stray impedances of the module itself. Researchers have also presented the notion that IGBT high-frequency oscillations occur in different types of SCs (i.e., type 1 and type 2) [16,17]. Although their work highlighted the factors that could trigger oscillations, internal module layout, IGBT characteristics, and application conditions, it provided very few details to aid the reader in understanding or suppressing the oscillations. Researchers have also investigated the plasma extraction transit time (PETT) effect as an excitation mechanism for high-frequency oscillations during the turn-off of paralleled IGBT chips [16,17], and some studies have discussed a combination of high-frequency oscillation source mechanisms, such as the dynamic impact ionization transit time (IMPATT) along with the PETT effect, during IGBT turn-off [18,19]. Unfortunately, no definitive interpretation of the mechanism leading to gate voltage oscillations that occur during the SC period has been provided to date.

Since gate voltage oscillations are one of the causes of power module failure, it is very important to understand the mechanism of gate oscillations under SC conditions and establish corresponding methods to suppress these oscillations. In this paper, we investigate this issue for automotive-grade multi-chip parallel power modules using two commercial power modules. We tested and analyzed one module that contained two chips and another module with only a single chip, and we observed considerably different behavior in the gate voltage oscillations between the two modules. The multi-chip module, with a parallel chip configuration, exhibited more severe oscillations than the single-chip module with an oscillation frequency that was several tens of megahertz. Therefore, in this study, we focused on the gate voltage oscillations in modules with parallel IGBT chips. To study the mechanism of these oscillations, we conducted a small-signal analysis of the parallel circuit, and the analysis yielded effective methods to suppress gate voltage oscillations. We also analyzed the two mainstream suppression schemes: increasing gate drive resistance and placing the drive capacitors in parallel. As a result of our investigation, we propose a simple and effective solution that involves parallel connections between the chips to mitigate gate voltage oscillations. This work contributes to the understanding of gate voltage oscillations in IGBT devices and provides a novel solution to enhance device performance during SCs.

The remainder of the article is organized as follows: Section 2 describes the setup of the short-circuit testing platform and the gate oscillation phenomena observed during short-circuit testing. Section 3 analyzes the mechanisms behind short-circuit gate oscillations. Section 4 investigates methods for controlling gate oscillations during short circuits. Finally, Section 5 concludes the paper.

## 2. Gate Oscillation Experiments under Short-Circuit Conditions

### 2.1. Construction of Short-Circuit Experimental Platform

Two three-phase full-bridge automotive-grade power modules with upper and lower bridge arms were tested, and the structural diagrams of the two modules are shown in Figure 1a. Internally, the modules consisted of three phases, *U*, *W*, and *V*, with the same chip selection and layout for each phase and two bridge arms, upper and lower, per phase. In Module A, each bridge arm consisted of two sets of chips, each rated for 750 V and 300 A, in parallel. In contrast, Module B had only one chip, also rated for 750 V and 300 A, per bridge arm. For simplicity, we focused our study on SC testing of the upper and lower bridge arms of the *U* phase with the module test temperature set at 25 °C.

A schematic of the test platform, powered by a high-voltage DC source, is shown in Figure 1b, where *U_dc_* represents the DC bus voltage, *C_dc_* represents the bus capacitance, and *L_s_* represents the stray inductance of the test system. To ensure low stray inductance in the DC bus during SC testing, the upper half-bridge, *M_U_*, of the device was used as the accompanying test bridge arm, and the lower half-bridge, *M_L_*, was used as the bridge arm under test, or device under test (DUT). A constant 15 V DC signal was applied to the gate of *M_U_* to keep it always on, and an SC pulse, with a pulse duration of *t_sc_*, along with a series drive resistance of *R_g_* was applied to the gate of *M_U_* During the test, the gate-emitter voltage, *V_ge_*, collector–emitter voltage, *V_ce_*, and SC current, *I_csc_*, of *M_U_* were simultaneously measured along with *U_dc_*. The assembled power module SC test platform with a *C_dc_* of 800 µF is shown in Figure 1c. The *L_s_* of the system was 50 nH, and a comprehensive list of equipment used in the test platform is provided in Table 1.

### 2.2. Gate Oscillation Experimental Test Results

Figure 2 shows the typical U-phase waveforms of *V_ge_*, *V_ce_*, and *I_csc_* in the Module A lower bridge arm under SC conditions. The plots in Figure 2a–f show waveforms with a bus voltage (*V_cc_*) ranging from 150 to 400 V in 50-V steps. After the IGBT was turned on, *I_csc_* rapidly increased, while *Vce* dropped in magnitude. As *I_csc_* increased, *V_ce_* gradually increased and *V_ge_* eventually exceeded the gate voltage level. This was caused by the displacement current flowing through the Miller capacitance. Due to the rise in *V_ge_*, an extremely high current flowed through the module, and when *V_ce_* approached the *V_cc_* level, the current became nearly constant (i.e., the device was in the active region). With *V_cc_* < 250 V, gate voltage oscillations did not occur, but with *V_cc_* ≥ 250 V, gate oscillations were present. This was because the LC oscillation circuit formed by the internal parasitic inductance and capacitance in Module A was more easily excited with high *V_ce_*, facilitating gate oscillations.

The SC test results for Module B with *V_cc_* = 400 V are shown in Figure 3. Like Module A, Module B exhibited an increase in *V_ge_*, but unlike Module A, *V_ge_* for Module B did not oscillate.

To investigate the gate oscillation mechanism, we disassembled the module and removed the encapsulation. The gate-emitter voltages of each IGBT chip (*V_ge_*_1_ and *V_ge_*_2_) were measured near their respective gate locations, and the results are shown in Figure 4. We observed that *V_ge_* for the two parallel chips exhibited a 180° phase shift in their voltage waveforms during oscillation, indicating that the gate oscillation occurs between the two parallel chips.

## 3. Gate Oscillation Mechanism Analysis

The results in the previous section showed that oscillations in *V_ge_* for the two-chip parallel module occurred when the IGBT was in the active region. Note that in the active region, the gain of an IGBT (*dI_c_*/*dV_ge_*) is higher than it is in the saturation region. To theoretically investigate the IGBT gate oscillation mechanism, we treated the parallel circuit of the two IGBT chips in the lower bridge arm as a feedback amplifier and used small signal analysis to simulate the circuit gain.

During the device switching process, the collector inductance and the emitter inductance together formed the power loop inductance, while the drive loop inductance was composed of the emitter inductance and the gate inductance. Depending on the switching stage, the effects of parasitic inductances on gate oscillations in the drive loop and power loop could be studied separately. Since the tested power module was a balanced three-phase device, only the upper and lower bridge arms of the U phase were considered, and their equivalent circuit model was established in LT-spice. Figure 5 shows the equivalent circuit of Module A under SC conditions. In Figure 5, *Q*_1_, *Q*_2_, *Q*_3_, and *Q*_4_ represent the parallel IGBT chips in the upper and lower bridge arms of module A, and *D*_1_, *D*_2_, *D*_3_, and *D*_4_ represent the parallel fast recovery diode (FRD) chips in the upper and lower bridge arms of module A. Figure 5 also shows the parasitic inductance, *L_cx_*, and resistance, *R_cx_*, of the collector loop, the parasitic inductance, *L_gx_*, and resistance, *R_gx_*, of the gate loop, and the parasitic inductance, *L_ex_*, and resistance, *R_ex_*, of the emitter loop. Note that the *x* in the subscript for each inductance and resistance parameter corresponds to the respective IGBT chip number, and *R_g_* with no additional subscript represents the gate drive resistance. Also, recall that *L_s_* represents the stray inductance of the test system. To analyze the small-signal circuit gain, an AC signal, *V*_1_, was applied to the IGBT chips *Q*_3_ and *Q*_4_, as shown in Figure 5. We then detected the amplitude and phase of the gate-emitter voltage *V_ge_*_3_ with *V*_1_ applied to *Q*_3_ and *Q*_4_, and we found that circuit oscillations occurred when
(1)$$\begin{array}{l} R_{e}(V_{ge3}/V_{1})\geq 1\;(\mathrm{amplitude}\;\mathrm{condition})\\ I_{m}(V_{ge3}/V_{1})=0\;(\mathrm{frequency}\;\mathrm{condition}) \end{array}$$

We simulated the equivalent circuit of Figure 5 in LT-spice and monitored the amplitude and phase of *V_ge_*_3_ with *V*_1_ applied to Q_3_ and Q_4_ and different values of *L_ex_* and *R_g_*. The simulated *L_ex_* values were set to 5, 15, and 25 nH, and the resulting amplitude and phase of *V_ge_*_3_ with *V*_1_ swept from 1 MHz to 1 GHz are plotted in Figure 6.

In the low-frequency area (*V*_1_ < 3 MHz), the phase of *V_ge_*_3_ was roughly 200°, but in the high-frequency area (*V*_1_ > 300 MHz), the phase of *V_ge_*_3_ was roughly 90°. We observed a rapid phase change near 80 MHz, and with an *L_ex_* of 15 or 25 nH, the phase reached 0°. At this point, the amplitude of *V_ge_*_3_ was greater than 0 dBV, and the oscillation conditions outlined in (1) were met, allowing oscillation to occur.

Since the *R_g_* and *L_ex_* of each IGBT chip are relatively easy to change compared to the other parameters, we studied the relationship between oscillation conditions and the values of *R_g_* and *L_ex_*. Figure 7 shows the simulated stable and oscillation regions for gate voltage with different values for *R_g_* and *L_ex_*. From the analysis, we concluded that high resistance between the gates of two paralleled chips and low inductance between the emitters could effectively suppress gate voltage oscillation. Additionally, Figure 7 can be used to choose appropriate values of *R_g_* and *L_ex_* to avoid IGBT gate oscillations.

To validate our theory and small-signal simulation results, a finite element model of Module A was developed in ANSYS Q3D software to extract the corresponding emitter parasitic parameters. The IGBT chip used in the module was rated for 750 V and 300 A with a typical turn-on time of 100 ns at 25 °C for *V_CE_* = 400 V, *I_c_* = 300 A, *R_g_* = 6 Ω, and *V_GE_* = ±15 V. Considering the impact of different drive resistances and DC bus voltages on turn-on time under practical working conditions, the simulation frequency range was set from 1 MHz to 10 MHz. Table 2 shows the extracted parasitic inductance values at 10 MHz, where the extracted *L_e_*_3_ is shown as 11.34 nH. This value, along with all other extracted *L_ex_* values, lies in the oscillation region identified in Figure 7, indicating the potential for oscillation.

## 4. Short-Circuit Gate Oscillation Suppression Methods

Common methods for SC gate oscillation mitigation include adjusting the gate drive resistance, adjusting the gate capacitance, and optimizing the power module’s internal layout to adjust parasitic inductance. In this section, we analyze the effects of these three methods individually.

### 4.1. Impact of Gate Drive Resistance on Gate Oscillations

To study the impact of different gate drive resistances on SC gate oscillations, SC tests were conducted with different values of *R_g_*, and the results are shown in Figure 8. Figure 8a shows that with *R_g_* = 10 Ω and *V_CC_* = 400 V, the gate experienced severe oscillations. In this case, *I_csc_* peaked at 5797 A, approximately 9.6 times the rated current. However, Figure 8 also shows that as the gate drive resistance increased, the gate oscillations gradually weakened. When the drive resistance reached 70 Ω, the gate oscillation phenomenon disappeared, but the *I_csc_* peak current was 5033 A, approximately 8.38 times the rated current.

We conclude that as the gate drive resistance increased, *d_i_*/*d_t_* gradually decreased, the turn-on became slower, and the loss increased. We further conclude that since *R_g_* significantly affects the switching characteristics of the power module, adjusting the value of *R_g_* solely for gate oscillations is not an ideal solution.

### 4.2. Impact of Gate Drive Capacitance on Gate Oscillations

In addition to adjusting the gate drive resistance, another common method to suppress gate oscillations is to increase the gate capacitance *C_g_*. To study the impact of different gate capacitance values on gate oscillations, we conducted tests by placing different capacitors in parallel between the gate and emitter. Figure 9a shows that with *V_cc_* = 400 V, *R_g_* = 10 Ω, and *C_g_* = 30 nF, the peak *I_csc_* was 5797 A, roughly 9 times the rated current. As *C_g_* increased, the oscillation of the gate-emitter voltage gradually weakened, and when *C_g_* increased to 400 nF, the gate oscillation disappeared. Unfortunately, we also observed a reduction in *V_ge_* with *C_g_* = 400 nF. Like the gate resistance, the gate capacitance has a significant impact on the switching characteristics of the power module and is not an optimal solution to mitigate gate voltage oscillations.

### 4.3. Impact of Adding Parallel Lines to the Common Emitter on Gate Oscillation

From the analysis in the previous sections, the parasitic emitter inductance has a significant impact on gate oscillation. Therefore, we used direct parallel bond wires on the surface of the two chip emitters to reduce the stray inductance between the emitters of each chip. The two IGBT chips connected in parallel were placed side by side, but the common emitter lines connected to the power terminals were relatively long. Therefore, we believed that directly bonding parallel wires on the emitters of the two parallel IGBT chips could effectively reduce the inductance between the two emitters.

The initial power module structure is shown in Figure 10a and the optimized structure with parallel bond wires on the common emitter of two parallel chips is shown in Figure 10b. The emitter inductance of the structure with parallel bond wires was extracted using ANSYS Q3D, and the values are shown in Table 3. With parallel bond wires and a gate resistance of 10 Ω, the extracted parasitic emitter inductance remained in the stable zone to prevent oscillation. To verify our approach, we prepared samples with parallel emitter bond wires and performed SC tests at bus voltages of 250, 300, 350, and 400 V. The oscillatory behavior of the gate-emitter voltage was significantly improved compared to the module without parallel bond wires, as shown in Figure 11. In addition to mitigating gate voltage oscillation, our method preserved the switching performance of the power module, highlighting the practical utility of our approach.

## 5. Conclusions

This study focused on the gate voltage oscillation of automotive-grade multi-chip parallel power modules under SC conditions. We performed SC tests on multi-chip parallel automotive power modules and single-chip automotive power modules. We observed gate voltage oscillation in the automotive power modules containing two chips when the IGBTs were in the active region, but we did not find oscillatory behavior in the single-chip modules. For the oscillatory behavior in the multi-chip modules, we also observed a 180° phase shift in the gate voltage of parallel chips.

Using small-signal analysis, we analyzed the mechanism of gate oscillation in multi-chip parallel modules, and under SC conditions, we found that the gate voltage oscillation in IGBT modules with two parallel chips was qualitatively described using a feedback oscillator model. Our analysis also indicated that a high resistive impedance between the gates of the two chips could suppress gate voltage oscillation by limiting the electrical energy between the emitters.

We experimentally investigated two common industry methods for suppressing gate oscillation, and we found that the gate drive resistance, *R_g_*, and gate capacitance, *C_g_*, have a significant impact on the module switching characteristics. Therefore, *R_g_* and *C_g_* modification is not an ideal oscillation mitigation method. As an alternative, we proposed a simple and efficient solution involving parallel bond wires on the common emitter terminals. By placing the bond wires in parallel, we significantly reduced the parasitic inductance of the emitter and effectively suppressed the gate voltage oscillation. Furthermore, this method did not degrade the switching characteristics of the power module, highlighting the practical utility of this simple approach.

## Figures and Tables

**Figure 1 sensors-24-02858-f001:**
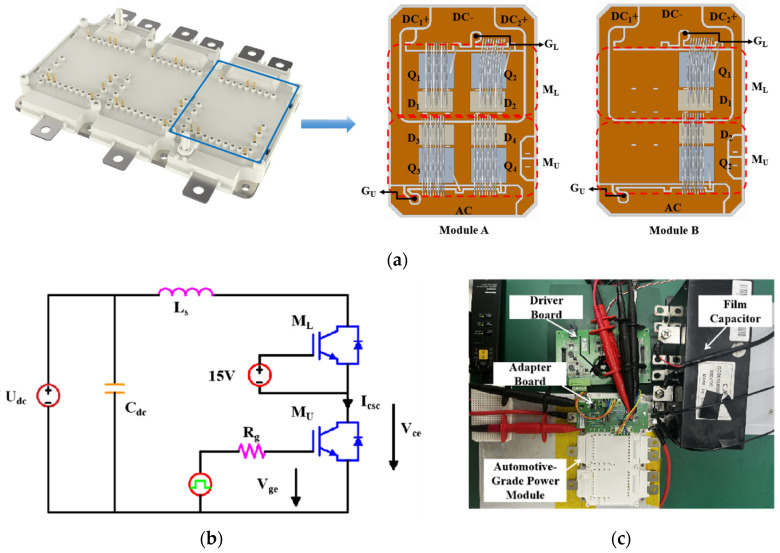
Automotive-grade power module and short-circuit test diagram. (**a**) Structural diagram of the automotive-grade power module and the internal structure of one phase. (**b**) Schematic diagram of the short-circuit test platform. (**c**) Photograph of the short-circuit test platform.

**Figure 2 sensors-24-02858-f002:**
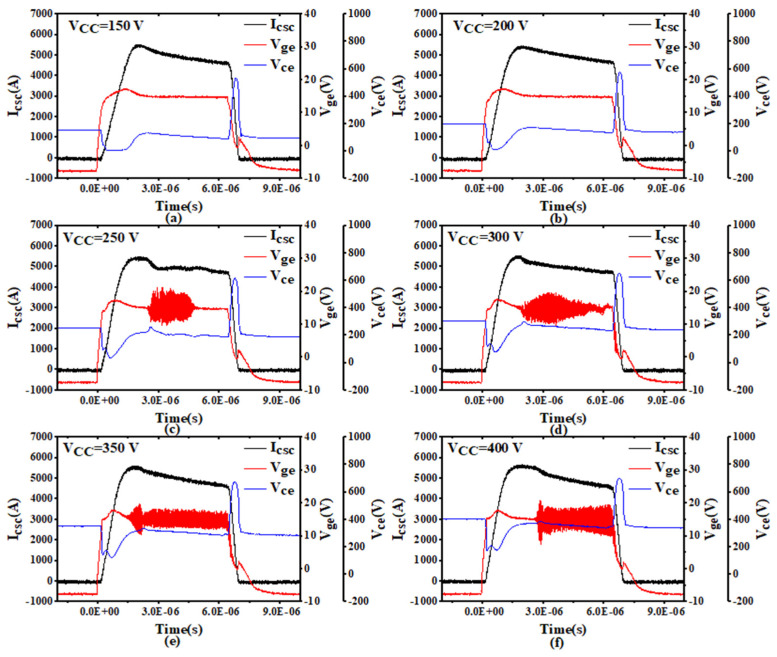
Short-circuit test results for Module A with DC bus voltages of (**a**) *V_cc_* = 150 V, (**b**) *V_cc_* = 200 V, (**c**) *V_cc_* = 250 V, (**d**) *V_cc_* = 300 V, (**e**) *V_cc_* = 350 V, and (**f**) *V_cc_* = 400 V.

**Figure 3 sensors-24-02858-f003:**
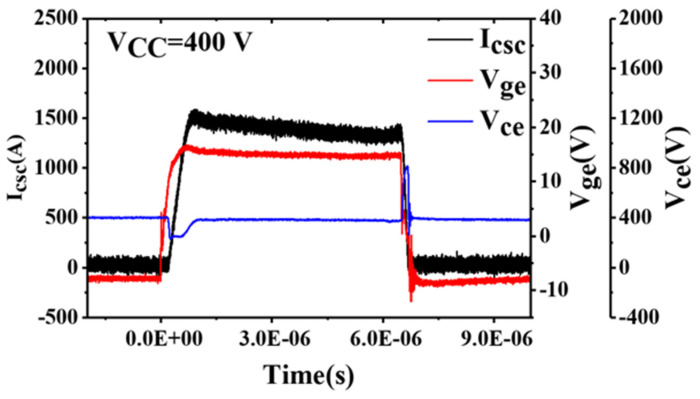
Short-circuit test results for Module B with *V_cc_* = 400 V.

**Figure 4 sensors-24-02858-f004:**
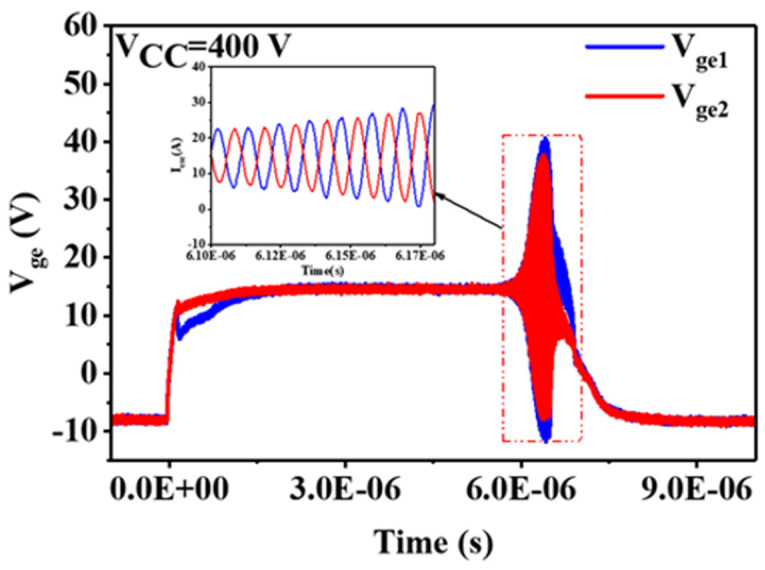
Module A gate oscillation waveforms for *V_ge_*_1_ and *V_ge_*_2_ during the short-circuit test.

**Figure 5 sensors-24-02858-f005:**
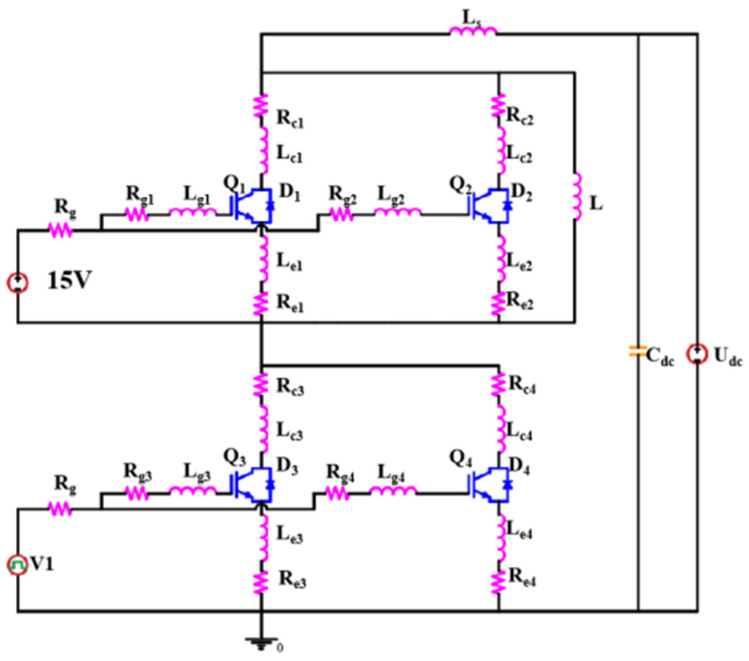
Equivalent circuit model of a multi-chip automotive-grade power module considering parasitic effects.

**Figure 6 sensors-24-02858-f006:**
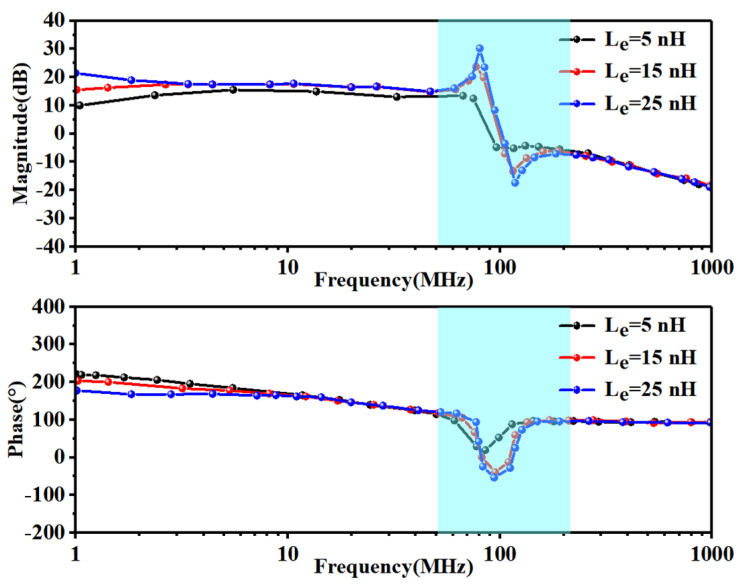
Amplitude and phase of *V_ge_*_3_ with *V*_1_ applied to *Q*_3_ and *Q*_4_ and different values for *L_ex_*.

**Figure 7 sensors-24-02858-f007:**
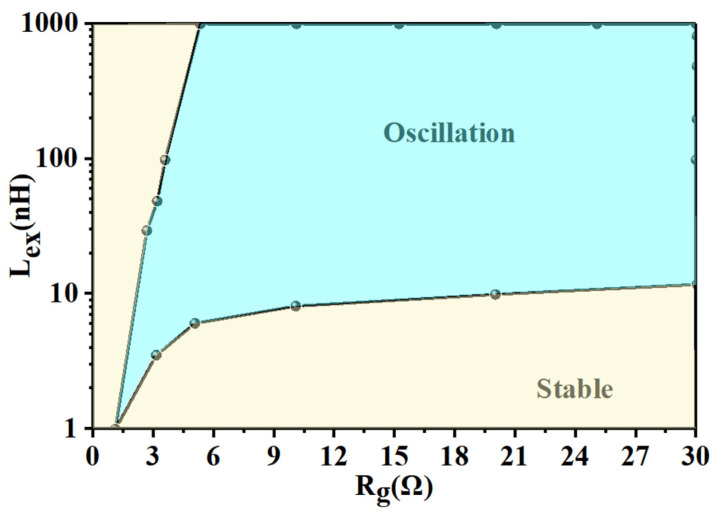
Stable and oscillation regions obtained through small-signal analysis.

**Figure 8 sensors-24-02858-f008:**
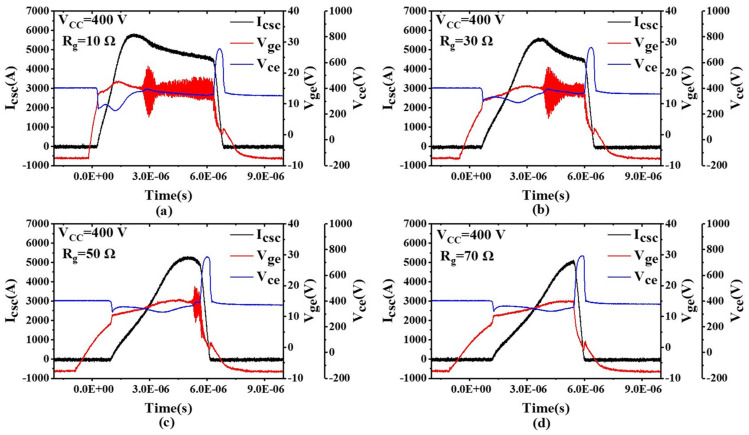
Short-circuit test results with different values of *R_g_*. (**a**) *R_g_* = 10 Ω; (**b**) *R_g_* = 30 Ω; (**c**) *R_g_* = 50 Ω; (**d**) *R_g_* = 70 Ω.

**Figure 9 sensors-24-02858-f009:**
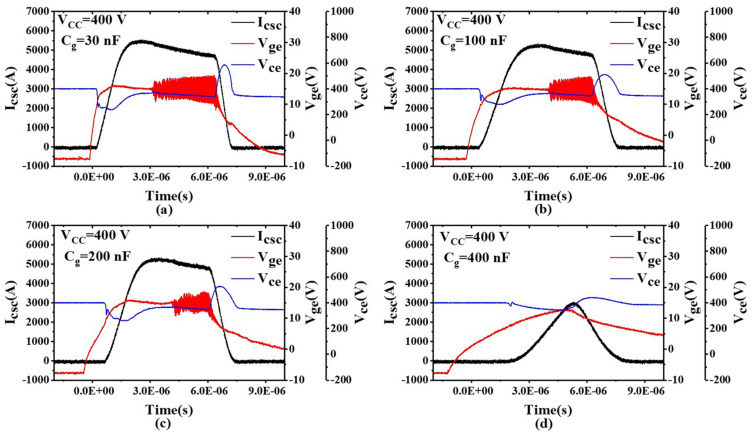
Short-circuit test results with different values of *C_g_*. (**a**) *C_g_* = 30 nF; (**b**) *C_g_* = 100 nF; (**c**) *C_g_* = 200 nF; (**d**) *C_g_* = 400 nF.

**Figure 10 sensors-24-02858-f010:**
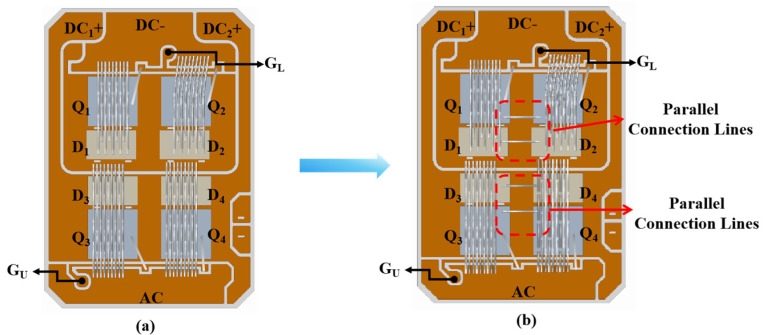
Structure diagram before and after the bond wire optimization method. (**a**) Initial structure. (**b**) Common emitter with parallel bond wires.

**Figure 11 sensors-24-02858-f011:**
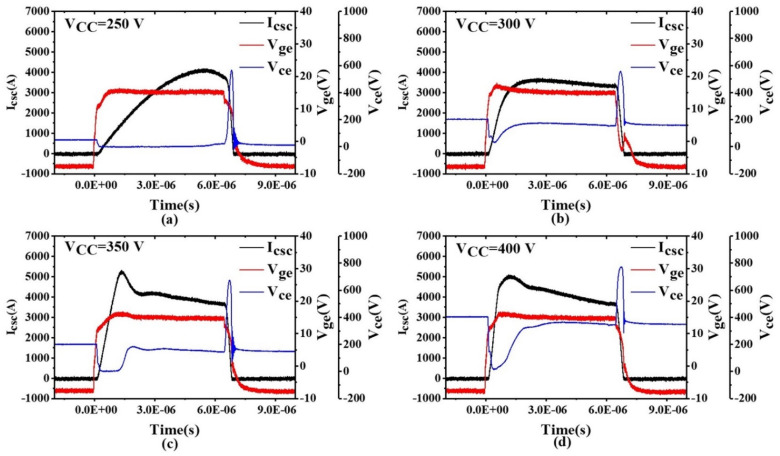
Short-circuit test results after adding parallel bond wires to the common emitter with bus voltages of (**a**) *V_cc_* = 250 V, (**b**) *V_cc_* = 300 V, (**c**) *V_cc_* = 350 V, and (**d**) *V_cc_* = 400 V.

**Table 1 sensors-24-02858-t001:** Short-circuit test platform experimental equipment and model numbers.

Equipment	Model
Oscilloscope	MSO56 5-BW-500
Pulse Generator	QTJ15610A
High-Voltage DC Power Supply	TDK Z+ 650-1
Auxiliary DC Power Supply	ITECH IT6302
High-Voltage Probe	Tek THDP0200
Rogowski Coil	IWATSU SS-286A
Load Inductance	FS-L-500
Driver Board	FZ1200R33KF2C
Adapter Board	Self-developed Adapter Board

**Table 2 sensors-24-02858-t002:** Extracted parasitic inductance of the lower bridge arm at 10 MHz.

Parameters	Values
*L_e_* _1_	10.76 nH
*L_e_* _2_	10.87 nH
*L_e_* _3_	11.34 nH
*L_e_* _4_	10.99 nH

**Table 3 sensors-24-02858-t003:** Emitter inductance of the initial structure and the optimized structure with parallel bond wires.

Parameters	Initial Structure	Optimized Structure with Parallel Bond Wires
*L_e_* _1_	10.76 nH	7.18 nH
*L_e_* _2_	10.87 nH	7.20 nH
*L_e_* _3_	11.34 nH	7.32 nH
*L_e_* _4_	10.99 nH	7.41 nH

## Data Availability

The data that support the findings of this study are available from the corresponding author upon reasonable request.

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
