# Peer review of "The Mechanism of Short-Circuit Oscillations in Automotive-Grade Multi-Chip Parallel Power Modules and an Effective Mitigation Approach"

_sensors, 2024, doi:10.3390/s24092858_

Round 1
Reviewer 1 Report
Comments and Suggestions for Authors
Dear Authors,
I recommend the following suggestions and observations be made:
In the summary there are no quantitative values of the results obtained and perhaps those that have been improved with the applied method.
They must present a comparative table of the results obtained in this work with at least 7 current works with the various methods used to reduce the gate oscillations of IGBTs. For example, by what percentage did efficiency and power density improve.
How many methods of suppressing gate oscillations are there?
Regarding the bibliography (references), some are more than 10 years old and 20 years old, so they must be replaced with more current references that are at least 5 years old. This is because they rely on it in their work and the conditions of the implementation and manufacturing technologies of the devices and PCB are different from more than 10 years than currently.
In line 57 they use references 8 and 9, which must be replaced by current works due to the technological change in the manufacturing of devices, integrated circuits and PCBs, so to speak, high-frequency working conditions should be considered more if the PCBs are multilayer.
Line 59-60: references 8, 19, 15 change them the same.
Line 63: reference 16 more than 20 years from 2004.
Line 70: References 18 and 19.
At the end of the introduction they must describe each section that makes up the article presented.
Line 111: Replace Ml with Mu.
Line 115: The inductance must be in henries 50 nH if applicable.
Line 121: Which IGBT transistor?
All parasitic effects of the IGBT device, interconnection lines and PCB traces must be considered. Because working at high frequencies these appear and degrade the integrity of the signals, unless protective guard rings have been placed and grounded to ground, this would improve operation and reduce gate oscillations.
Remember that IGBTs are voltage-controlled devices, which causes the capacitors to change value or disappear or become present depending on the operating region and the conditions of the gate and the voltage applied in CE.
Figure 6 shows that there are peaks in the range of 60 to 80 MHz, these peaks represent oscillations in that range, that is where the problem must be attacked, given that the parasites resonate in that range and are a product of positive local oscillation. And that is not clarified in the work, they must work on the small signal model replacing the IGBT model and obtain representative equations. And there aren't any.
Author Response
Dear editor and reviewer,
We are very pleased to have learned from your comments regarding our manuscript titled “The Mechanism of Short-Circuit Oscillations in Automotive-Grade Multi-Chip Parallel Power Modules and an Effective Mitigation Approach”. We sincerely appreciate your careful reading and thoughtful comments and valuable suggestions. We have carefully considered the comments and revised the manuscript accordingly.
Our point-by-point responses can be found in the attachment. Relevant changes have been made in the original manuscript according to the Reviewers’ comments.

Reviewer 2 Report
Comments and Suggestions for Authors
The paper is largely experiment-driven, so it can represent a useful reference for practitioners in the field. Some insight is proposed in sec. 4.3 highlighting the role of the parasitic emitter inductance in the onset of oscillations, confirmed by the tests conducted.
Some typos concerning the units of measurement should be corrected, i.e.:
- p. 3/12, line 115: 50 nF --> 50 nH
- p. 7/12, line 185: 0 dB --> 0 dBV
In Fig. 7, I suggest replacing L_e with L_ex for consistency with the notation in the main text.
Finally, I would improve Fig. 10, which I find rather difficult to read.
Author Response

(The authors gave the same response as above.)
